# Methodological and Conceptual Progresses in Studies on the Latent Tracks in PADC

**DOI:** 10.3390/polym13162665

**Published:** 2021-08-10

**Authors:** Tomoya Yamauchi, Masato Kanasaki, Rémi Barillon

**Affiliations:** 1Graduate School of Maritime Sciences, Kobe University, Hyogo 657-8501, Japan; kanasaki@maritime.kobe-u.ac.jp; 2Institut Pluridisiplinaire Hubert Curien, UMR 7178, CNRS, The University of Strasbourg, 67081 Strasbourg, France; remi.barillon@iphc.cnrs.fr

**Keywords:** PADC, poly (allyl diglycol carbonate), latent track, track core radius, *G*–value, layered structure, *REFIT*, *NISE*, detection threshold, chemical criterion

## Abstract

Modified structure along latent tracks and track formation process have been investigated in poly (allyl diglycol carbonate), PADC, which is well recognized as a sensitive etched track detector. This knowledge is essential to develop novel detectors with improved track registration property. The track structures of protons and heavy ions (He, C, Ne, Ar, Fe, Kr and Xe) have been examined by means of FT-IR spectrometry, covering the stopping power region between 1.2 to 12,000 eV/nm. Through a set of experiments on low-*LET* radiations—such as gamma ray-, multi-step damage process by electron hits was confirmed in the radiation-sensitive parts of the PADC repeat-unit. From this result, we unveiled for the first-time the layered structure in tracks, in relation with the number of secondary electrons. We also proved that the etch pit was formed when at least two repeat-units were destroyed along the track radial direction. To evaluate the number of secondary electrons around the tracks, a series of numerical simulations were performed with Geant4-DNA. Therefore, we are proposing new physical criterions to describe the detection thresholds. Furthermore, we propose a present issue of the definition of detection threshold for semi-relativistic C ions. Additionally, as a possible chemical criterion, formation density of hydroxyl group is suggested to express the response of PADC.

## 1. Introduction

### 1.1. Background and Applications

More than 40 years have passed since the re-discovery of poly(allyl diglycol carbonate), PADC, as a polymeric etched track detector, which had been developed as an optical plastic material [1]. PADC detectors are best known under the trade name of CR-39. In etched track detectors and solid-state nuclear track detectors applications, PADC detectors are chemically etched in alkaline solution of NaOH or KOH to enlarge latent tracks of proton and heavy ions to etch pits that are observable under optical microscopes. From the geometry of each etch-pit, it is possible to identify the nuclear species and to evaluate the incident energy. A more detailed description of nuclear track detectors, including fundamental aspects and their applications, can be found in *Handbook of Radioactivity Analysis* [2].

The track registration properties of PADC detectors are hardly affected by low-*LET* (Linear Energy Transfer), radiations, electrons, X rays and gamma rays, at absorbed doses below about 10 kGy. This property has been used, in various branches of science and technology related to mixed radiation fields, for experiments on the inertial confinement fusion for instance [3], the diagnosis of ion beams from intense-laser plasma [4,5,6,7], dosimetry of space radiations [8,9,10], and the evaluation of local dose distribution in boron neutron capture therapy (BNCT) [11] notably. Despite a breadth of applications, a decade ago the fundamental mechanisms of latent track formation in PADCs remained to be explained. From this knowledge, the authors are confident that this new-knowledge on latent track structure and track formation process in PADC will give us the essential knowledge to develop novel detectors with better track registration property, for novel and improved applications.

### 1.2. One Decade of Methodological and Conceptual Evolution

Two PhD studies have been performed and their progresses are the body of the present contribution [12,13]. For the two research projects, the Fourier transform infrared (FT-IR) spectrometry has been the main experimental method. We have succeeded in making quantitative analyses of track structures, as well as qualitative observations. Our key-method to perform the quantitative analyses has been the production of thin films of PADC, which allow us to reach unsaturated IR spectra (in the methodology section, we provide the details on how to produce the thin PADC film). The first quantitative experiment was carried out at the CIMAP-GANIL (France) where on-line FT-IR measurements were available, using the 15 µm thickness films [14]. Following on these experiments we developed a thinning process of PADC films to reduce the thickness to less than 3 µm, with a precise and accurate estimation of their thickness [15]. Thanks to this new material, off-line measurements of protons and heavy ions (He, C, Ne, Ar, Fe, Kr and Xe) were made, covering the stopping powers ranging from 1.2 to 12,000 eV/nm [15,16,17,18,19,20,21,22,23], as well as low-*LET* radiations [20,21,24,25,26]. As a reference, effects on exposure to ultraviolet (UV) photons with a wavelength of 222 nm were also executed [27]. Additionally, as quantitative analyses, we have evaluated the linear damage density, which is the number of losses of considered functional groups per unit distance of track length and effective track core radius. From the loss in the considered chemical groups, we obtained the radiation chemical yield (G value) for each functional group. From these chemical damage parameters, we could then express and understand the latent track structure (we are also giving adequate explanation on how to obtain these values from changes in IR spectra due to the exposures).

Based on the evaluation of the effective track core radius, we found layered structure with three different regions in the radial direction of the latent tracks [18,23]. As a polymer network, PADC consists of two different sections with radiation-sensitive parts and relatively radiation-tolerant ones (Figure 1). The former is made of separated linear parts with a length of about 1.5 nm, which are sets of single ether bond with two carbonate esters. The latter has a “polyethylene-like” structure formed during the polymerization, working as a frame for the PADC.

After the exposure to energetic heavy ions, in the most outer region, only ether is lost. In the middle, ether and carbonate ester are damaged. In the central region, parts of the polyethylene-like structure are also modified. Through the experiments using low-*LET* radiations of 28 MeV electron beams [25], gamma rays from ^60^Co and 1.5 keV ultra-soft X ray [26], it was found that a single electron can cleave the ether, but it cannot break the carbonate ester. Carbonate ester turns out to be radiation-sensitive when the adjacent ether is lost. We have called the dose above which carbonate ester becomes radiation-sensitive the “critical dose”. These results from low-*LET* radiations are concordant with the layered structure of ion tracks. The number of secondary electrons should determine the damage level in each region. However, it was well demonstrated that CO_2_ is produced in PADC during exposures to ions and low-*LET* radiations [28,29,30,31,32,33], which is attributable to the de-carbonyl reaction of the carbonate esters. These observations suggest that the parts between the two carbonate ester bonds in each repeat-unit can be segmented into two CO_2_ and other small molecules as gaseous components along the nuclear tracks [14] that will leave a stable damage where the radiation-sensitive parts should be totally or partially lost. Through the etching tests, we have obtained the response of PADC as an etch track detector and determined the detection thresholds for protons and He and C ions [34]. It turned out that at least two radiation-sensitive parts should be destroyed in the radial direction of the ion tracks to reach the detection threshold. In this stage, it becomes possible to discuss the relationship between the response and the latent track structure. It was strongly suggested that we should estimate the number of secondary electrons around ion trajectory to obtain the total view of damage in PADC and the relation with the detection threshold. As a physical model of tracks in condensed materials, we have paid a special attention to the local dose distribution model started by Katz [35]. The radial structure of latent tracks in PADC was also discussed based on the dose distribution model [36,37]. We further the physical analysis with numerical simulation using the Monte Carlo simulation code Geant4-DNA, which has been developed to model radial energy deposition around ion track [38]. In its present state, the simulation is able to reproduce dose distribution comparable to the Katz model in water. From these simulations, new physical parameters have been proposed by Kusumoto et al. to describe the response of PADC detector [39,40]. Radial Electron Fluence around Ion Tracks, *REFIT*, has been defined as the number density of secondary electron passing through the cylinder with identical track-axis as the ion-pass [39]. The Number of Interactions by Secondary Electron, *NISE*, has in turn been defined as the number of interactions counted in a series of cylinders coaxial to the ion trajectory [40]. These physical parameters are clear improvement in comparison with the traditional restricted energy loss, *REL* [41] for instance.

After discussing the physical criterions, we are now turning to the chemical aspects and propose a chemical criterion to describe the response of PADC, based on the quantitative evaluation about formation density of hydroxyl groups along ion track [22]. After exposure, the density of each functional group composing a PADC is generally reduced, while one can find two emerging peaks of CO_2_ gas and of hydroxyl group [33]. The hydroxyl groups are produced at the new end-points where the damage occurred. As the hydroxyl group is a typical hydrophilic group, its density should govern the penetration velocity of the etching solution, which should be identical to the track etching velocity [42].

Provided with this framework, the contribution focuses on (1) the experimental and (2) the practical aspects of a decade of research, and (3) a discussion of the forthcoming issues and the directions we foresee research on latent tracks in PADC could take, for other research group can continue and reproduce this research. To complement the present study from the viewpoint of radiation chemistry in PADC, the reader is referred to the recent review by Fromm et al. [43].

## 2. Materials and Methods

### 2.1. Preparation of PADC Films

The first step is to obtain the thin films of PADC samples suited for FT-IR measurements. For this purpose, we have used commercially obtained BARYOTRAK sheets with a nominal thickness of 100 µm (Fukuvi Chemical Industry Co., Ltd., Fukui, Japan), which was produced from purified monomer (purity superior to 99.7%). The sheets were cut in a 20 × 20 mm^2^ size. In order to reduce the sample thickness, we have applied chemical etchings with three steps. The etchings were made in 6 M KOH solution kept at 70 °C without stirring. As a first step, the films were reduced from 100 µm to about 20 µm, and the samples were held using the stainless clips. It took about 40 h to reduce the thickness down to 20 µm. Handlings of thin films with a normal forceps became difficult when the thickness became less than 15 µm. For further etching, we utilized a stainless steel sieve as a sample holder (see Figure 2). The second step was started from 20 µm to the aimed thickness. Following each etching, the samples were rinsed in distilled water carefully and then dried in a clean and dark space after blowing out the excess water on the surface. Then we evaluate the thickness of the film based on the Beer–Lambert law and FT-IR measurements.

According to the Beer–Lambert law, the absorbance of considered functional group, *A*, is expressed by the product of the molar absorption coefficient, ε, in cm^−1^ M^−1^, molar concentration, *c*, in M and sample thickness, *t*, in cm, so that:(1)A=εct

Prior to film thinning process, we had confirmed the quantitative relation between the absorbance of IR peaks at 789 and 878 cm^−1^, which are assigned to CH rocking out-of-plane deformation, and the thickness of films ranging from 10 to 60 µm (Figure 3). The value of film thickness was calculated from the average of five different points from the four corners and the center of each film using a micrometer. Absorbance is well proportional to the thickness, as expected. The following Equations for the sample thickness in cm were derived by the least-squares method, for the peak at 789 cm^−1^ as:(2)$$t=4.55\times 10^{-3}A_{789{\mathrm{cm}}^{-1}}$$
and for the peak at 878 cm^−1^ as:(3)$$t=1.25\times 10^{-3}A_{878{\mathrm{cm}}^{-1}}$$

We estimated the film thickness using these Equations after each etching. It should be noted that we have evaluated the absorbance from the height of each peak. FT-IR measurements were performed using a spectrometer FT/IR-6100S (JASCO), which operates under a full vacuum with the interferometer, the photon-detector, and the sample room during the measurements. This set-up excludes the influence of moisture and carbon dioxide in the ambient air. This setup has proven to be necessary to obtain comparable and repeatable FT-IR spectrometry data.

The third step of etching has been designed to be the last. The thickness of the films was adjusted by controlling the etching time. After obtaining the required thickness, the films were rinsed in distilled water and dried, and then kept in a clean and dark space. FT-IR measurement was then carried out just before each irradiation experiment once again as a control-test.

### 2.2. FT-IR Spectra of PADC

Figure 1 illustrates a repeat-unit of PADC. It has an ether bond in the center and two carbonate ester bonds in the symmetric positions. In both directions from the center, two methylene groups (CH_2_), namely, the ethylene group, are sandwiched with the ether and the carbonate ester bonds. As described above, these are composing radiation-sensitive part in PADC. The other relatively radiation-tolerant parts are three-dimensional polyethylene-like structure, produced as a result of polymerization [44]. At each three-ways junction of the polymer network, one can find a methine group (CH). Chemical formula of PADC is C_12_H_18_O_7_. In each repeat-unit, two hydrogen atoms are working as methine group and other 16 hydrogen atoms belong to methylene group, and half of methylene group are in the radiation-sensitive parts. The density of PADC is 1.31 g/cm^3^.

From the result of the FT-IR spectra (Figure 4), PADC films with a thickness of 10 µm is saturated, while the 3 µm spectrum is unsaturated, and suitable for the research. So, we have utilized samples with thicknesses between 2 and 3 µm, to analyze the peaks corresponding to the ether, carbonate ester and CH (methylene and methine). The peak assignments have been reported by some key contributions in the literatures [45,46,47]. The three peaks at 1142, 1092 and 1028 cm^−1^ are assigned to stretching vibration of ether, indicated by lateral short bars in Figure 4. Usually, we chose to measure the third peak, which is relatively independent from others. Two strong absorption peaks have also been observed at 1750 and 1250 cm^−1^. The former is the peak of carbonyl (C=O) and the latter is that of C–O–C. These are composing the carbonate ester bonds. For the analyses of CH, we have chosen the fairly isolated peak at 789 cm^−1^, that was selected also for the use of film thickness evaluation as previously described. This peak corresponds to the sum of molar concentrations of methylene and methine. The thicker samples with 10 or 15 µm were used to estimate the peaks for hydroxyl group around 3500 cm^−1^. We must take the absorbed water into account to evaluate the amount of hydroxyl group formed in the polymer network. One can find three peaks around there, at 3640, 3540 and 3700 cm^−1^ [22,33,47] The first peak is assigned to the anti-symmetric vibration of water and the second is that for both the symmetric vibration of water and the hydroxyl group of the polymer. The third one is the first over tone of carbonyl. Therefore, we need the condition of vacuum to measure the concentration of the hydroxyl group of the polymer. The concentrations of hydroxyl group in PADC films increase monotonically due to the exposure to ionizing radiations [22,33,47]. We have derived the molar absorption coefficient as ε = 9.7 × 10^3^ M^−1^cm^−1^ for hydroxyl group [22]. This value was originally attained, using absorbed water in PADC. Recently, this value was ascertained to be valid for the hydroxyl group in poly(vinyl alcohol). When we use this value to obtain the concentration of hydroxyl group, we must adopt the peak area to assess the absorbance.

Protons and heavy ions irradiation were mainly performed at the port of the medium energy irradiation room of Heavy Ion Medical Accelerator in Chiba, HIMAC, a facility of the National Institute of Radiological Sciences, National Institutes for Quantum and Radiological Science and Technology (QST-NIRS), Japan. Incident energies were less than 6 MeV/u. This allowed us to investigate in the energy regions close to Bragg Peak [16,17,18,19,20,21,22]. The other port of biological irradiation room of HIMAC was used for heavy ions irradiation (C, Fe and Xe ions) with energy in excess of 100 MeV/u [12,20]. Protons irradiation energy with higher than 20, 30, 70 MeV were made at the C-8 port of the AVF-930 cyclotron in QST-NIRS [23]. In addition to these ion irradiation experiments, we carried out experiments with low-*LET* radiations. 28 MeV electron beams were irradiated at the L-band linear accelerator installed at The Institute of Scientific and Industrial Research, ISIR, Osaka University, Japan [13,28]. Gamma irradiations were also made at SRI, using intense Co-60 sources [12,13,21,24,26]. Examined absorbed doses were ranged from 10 to 1000 kGy. The local dose for proton tracks in PADC is within this dose range [15]. About the heavy ions, the local doses at the effective track core radius for the loss of carbonate bonds were about 1000 kGy [15].

## 3. Results and Discussion

### 3.1. Removal Cross-Section

Figure 5 shows the decreasing behavior of the relative absorbance $A/A_{0}$
of ether and carbonyl due to the exposure to 130 MeV/u Xe ions, against ion fluence, F. We found a linear reduction of the relative absorbance for both functional groups with fluence. The relative absorbance $A/A_{0}$ is the ratio of net absorbance of considered functional group after the irradiation A to the original absorbance $A_{0}$. The observed trends are well expressed by the following experimental formula against ion fluence:(4)$$A/A_{0}=1-\sigma_{i}\cdot F$$
where $\sigma_{i}$ is an experimental constant in units of cm^2^. This constant is the slope of the fitted line. Physically, it means the removal cross-section or the effective track core cross-section in which the considered group was lost. Before applying this type of formula to our data sets, we made preliminary discussions based on the track overlapping model [42]. From practical aspect, it may be possible to regard the overlapping is negligible, as far as we observe liner relation in the reduction behavior of the relative absorbance against ion fluence. We think, however, it is better to selectively utilize the data of relative absorbance that are greater than 0.9. We have observed similar decreasing trends of relative absorbance for ether, carbonate ester and CH, for all the examined cases of protons and heavy ions irradiations. Then we have obtained a set of data on the removal cross-section for each functional group, for each ion, with different incident energies, by the least-square relation fitted to Equation (4).

According to the Beer–Lambert law (cf. Equation (1)), absorbance is given as the product of molar absorption coefficient, ε, molar concentration, *c*, and sample thickness, *t*. The molar absorption coefficient and the sample thickness are not affected by the irradiations. Only the molar concentration should be modified as *c′*, from the original concentration of *c*_0_ due to the ion exposure. Namely, we have the following relation for each functional group before the irradiation as:(5)$$A_{0}=\varepsilon c_{0}t$$
and we have the next relation after the irradiation as:(6)$$A=\varepsilon c^{\prime }t$$
therefore, the relative absorbance becomes:(7)$$A/A_{0}=c^{\prime }/c_{0}$$
where we introduce the original density of a considered bond as $N_{0}$. Each bond corresponds to an IR peak. The survival fraction of the bond is given by $N\left(F\right)/N_{0}$, that is the ratio of density after the irradiation N(F), to that of the original one $N_{0}$. In the region of ion fluence, where the overlapping of tracks was negligible, the relative absorbance is equivalent to the survival fraction:(8)$$A/A_{0}=N\left(F\right)/N_{0}$$
namely, we obtain the simple relation as:(9)$$N\left(F\right)/N_{0}=1-\sigma_{i}\cdot F$$
with a unit fluence, it becomes as:(10)$$\sigma_{i}=\frac{N_{0}-N\left(1\right)}{N_{0}}$$
the removal cross-section is the loss ratio of the considered bond by a single ion track. We have applied the following chemical damage parameters to discuss the features of ion tracks, which were derived from the value of the removal cross-section.

### 3.2. Chemical Damage Parameters

The linear damage density, or the damage density, L, is defined as the number of losses of considered bonds per unit distance of the track length. We can obtain the value of the damage density as the product of the removal cross-section and the original density of the considered bond, defined as:(11)$$L=\sigma_{i}\cdot N_{0}$$
as a simple approximation, one can expect the damage density to be proportional to the stopping power or *LET* for each ion in PADC.

To express the radial size of the damage, the effective track core radius, $r_{t}$, has been applied. Actually, the network structure of PADC should be severely altered around the break points, including cross-linking and recombination resulted in the modified structure. When we think about the effective track core radius, it is based on the original configuration of the network. It is defined as the radius in which the considered chemical bonds are lost as in the original one. In other word, considered bonds are regards to be lost when the corresponding scissions occur in the order of the distance from the track center. We can attain the value from the removal cross-section using the next Equation:(12)$$r_{t}=\sqrt{\sigma_{i}/\pi }$$

Usually, the radiation chemical yield (G value) is determined based on the total absorbed dose of the considering material. In our studies on the latent ion tracks, we have utilized the next simple relation as:(13)$$G=L/\left(-\frac{dE}{dx}\right)$$
in this calculation, the damage density is simply divided by the stopping power in the films. Throughout our studies, the stopping power has been calculated using SRIM code [48]. As shown in Equation (13), the damage density is obtained as the product of the G value and the stopping power, and this product is proportional to the square of the effective track core radius for each functional group. One can derive G values from the effective track core radius and the corresponding stopping power, using these relations.

We shall consider the relationship among these three chemical damage parameters, in a simple case. Here, we assume that G value is independent of ion species and its energy. From Equation (13), the damage density is proportional to the stopping power. As expressed by Equation (11), the removal cross-section is proportional to the damage density. Therefore, the effective track core radius is proportional to the square root of the stopping power, according to Equation (12).

From the experimental results of PADC and other plastic materials, we have derived many fitted functions to express the effective track core radius as a function of the stopping power, in a form of the power function as:(14)$$r_{t}=\alpha {\left(-dE/dx\right)}^{\beta }$$
where α and β are the fitting parameters. We found the G value for the loss of carbonyl in bisphenol A polycarbonate, PC, was almost independent of the stopping power [17,19]. For the case of PC, the exponent β is almost 0.5. In other plastic of poly (ethylene terephthalate), PET, we found the increasing behavior of G value for the loss of carbonyl with increasing the stopping power [17]. In addition to this, we found the distinct step-like increase around the detection threshold of PET [49]. For PET films, β is greater than 0.5. Return to PADC, β is less than 0.5, for ether and carbonate ester at stopping powers less than 800 eV/nm. G values for losses of carbonyl and ether in PADC increase with decreasing the stopping power. This is true for protons and some heavy ions that we have examined [20,23]. This is one of the great features of PADC one cannot find in other polymers, which should make it possible to register etchable tracks of protons in it.

As another simple case, it was assumed that the core radius was independent of the stopping power. The damage density is also constant in this case. Therefor, G value decreases with the stopping power, according to Equation (13). We will find the similar case for the loss of CH at the stopping powers between 10 and 50 eV/nm, in the other section.

### 3.3. Critical Dose for Loss of Carbonate Ester

Contrary to the ions, low-*LET* radiations were hypothesized to produce spatially uniform damage. In our early studies on the effects of gamma rays from ^60^Co, we obtained G value of 20 scissions/100 eV for the loss of carbonate ester [24]. In the later, this was modified as that the value was valid above the critical dose of about 60 kGy [26]. Through the FT-IR studies on the effects of 28 MeV electron beams on PADC, applying the track overlapping model [42,50,51,52], it was found that the carbonate ester was radiation-sensitive after the damage of the ether in the same repeat-unit [25]. Below the critical dose, carbonate ester was hardly damaged by low-*LET* radiations, including 1.5 keV ultra-soft X rays [25]. Therefore, at least two hits by electrons are indispensable to break the carbonate ester bond in each repeat-unit. A similar multi-step damage formation process resulted in the loss of carbonyl in PADC and it was observed due to the exposures to UV photons with a wavelength of 222 nm (photon energy of 5.58 eV). It was indicated that the carbonyl begins to be cleaved, when 10% of total amount of ether and C–O–C is damaged. Yields for loss of carbonyl was fairly lower than those of ionizing radiations [27].

After the break of ether in the repeat-unit, entirely different types of thermal motions should start around the created open-ends. In order to understand the effect of thermal vibrations on the G values, we are conducting a new series of experiments at the beam port of CIMAP-GANIL, France, that is on-line FT-IR measurements at low temperatures (>11 K) [53].

### 3.4. Layered Structure of Ion Tracks

A series of FT-IR spectrometric studies has revealed that the latent tracks in PADC has their own layered structure in the radial direction [18,23]. As an example, the structure of 4.8 MeV/u C ion track is illustrated in Figure 6. This is an average depiction along track length. It has been measured that there are three different regions, as shown in Figure 6. The effective track core radius for loss of ether is 1.8 nm and that for loss of carbonate ester is 1.4 nm. Only ether is damaged at the outermost region between 1.4 and 1.8 nm. We think this region should be produced by single hits of secondary electrons, refereeing to the necessity of two hits for the loss of carbonate ester observed in low-*LET* radiations. In the middle region between 1.2 and 1.4 nm, ether and carbonate ester are lost. In the center region with in 1.2 nm, all functional groups are lost, including CH. The methylene groups in the radiation-sensitive part should be lost, when the ether and carbonate esters were damaged.

### 3.5. Effective Track Core Radius and Detection Threshold

We have used the empirical formula based on experimental data to express the effective track core radius for the loss of each functional group [23], which we approached from the concept of the stopping power (Figure 7). Results show that the radius for carbonyl and that for carbonate ester have identical meaning. For the radius for ether and that for carbonyl, we have applied two different segments in the stopping power divided at 800 eV/nm, to obtain the best fit. The first segment is below 800 eV, and the second one is above this value. At 800 eV/nm, the core radius to experience loss of ether is about 2 nm, which is equivalent to the length of the repeat-unit. This means that more than two repeat-units are damaged at the second segment in the radial direction.

For the effective core radius for loss of ether $r_{ether}$, we have derived the following Equations for each segment as:(15)$$r_{ether-1}=0.19{\left(-dE/dx\right)}^{0.35}$$
for the first segment and as:(16)$$r_{ether-2}=0.04{\left(-dE/dx\right)}^{0.59}$$
for the second segment, where $r_{ether-1}$ and $r_{ether-2}$ are in nm unit and the stopping power unit is in eV/nm. The exponent of Equation (15) is less than 0.5 and that of Equation (16) is larger than 0.5. As discussed in the previous section, the dependence of G valeus on the stopping power is opposite between the two segments. In the first segment, the G value decreases with increasing stopping power. In the second segment, G value increases with the stopping power.

The least square fittings for the effective core radius for loss of carbonyl, $r_{C=O}$, gave the following empirical relationship for the first segment:(17)$$r_{C=O-1}=0.14{\left(-dE/dx\right)}^{0.35}$$
and for the second section, it gives:(18)$$r_{C=O-2}=0.03{\left(-dE/dx\right)}^{0.60}$$
the radius of the loss of carbonyl is always smaller than that of the loss of ether with almost constant ratios for each segment (Figure 7). This trend is concordant with the previously described requirement to have at least two hits for the loss of carbonate ester to occur.

The trends of effective track core radius for loss of CH, $r_{\mathrm{CH}}$, are significantly different between the protons with energies of 20, 30, 70 MeV and the other heavy ions, as indicated in Figure 7 [23]. The core radius of the former is expressed as:(19)$$r_{\mathrm{CH}-a}=0.11{\left(-dE/dx\right)}^{0.47}$$
for the region of the stopping powers between 1 and 10 eV/nm. The power exponent of Equation (19) is close to 0.5. So, G value for loss of CH is almost independent of the stopping power. It is noteworthy that the G values for the protons are remarkably higher than those for the heavy ions around the Bragg peaks. We think that the losses of CH are limited within methylene located in the radiation-sensitive parts in each repeat-unit in this region. Then, all induced damage is supposed to occur within each repeat-unit. Experimentally obtained data for the heavy ions are well fitted by the following relation:(20)$$r_{\mathrm{CH}-b}=0.03{\left(-dE/dx\right)}^{0.59}$$
for the region of the stopping powers between 50 and 12,000 eV/nm. The component is greater than 0.5. So, G value increases gradually with increasing the stopping power.

At the stopping powers ranging from 10 to 50 eV/nm, the damage to CH hardly expands in radial direction, nevertheless the damage on ether and carbonate ester increases monotonically. The effective core size for loss of ether is ranging from 0.43 to 0.75 nm in this region. As the length of the repeat-unit is about 2 nm, two neighbors of ether have to be involved in the core radius. It was inferred that the damage of CH in the radiation-tolerant part start above the stopping power of 50 eV/nm. We think that damage at the radiation-tolerant part become possible after two adjacent radiation-sensitive parts were damaged. Now, we shall pay an attention on the detection thresholds for proton, He and C ions are within this region, which are indicating in Figure 7 as the short longitudinal bars. The thresholds for proton, He and C ions are 19, 37 and 39 eV/nm, respectively [34,39]. The value for C ion was modified from 55 eV/nm in resent experiments, as denoted in the following section.

We think that at least two radiation-sensitive parts should be destroyed in the radial direction of ion tracks to produce any etchable latent tracks in PADC. It becomes possible to discuss the relationship between the detection threshold and the latent track structure. These experimental results implied that we should estimate the number of secondary electrons around ion trajectory to obtain the total view of damage in PADC and the relation with the response.

### 3.6. Radial Electron Fluence around Ion Track

We have applied the local dose distribution model initiated by Katz [35], to discuss the radial structure of latent track. Experimentally ascertained size of the effective track core radius in polymeric materials has been discussed based on the local dose, including PADC [15,20,23,36,37,54,55,56,57,58]. The dose model has been available to make a quick check to each experimental result. However, we were facing the novel problem to evaluate the number of secondary electrons around ion tracks to explain the physical base of the detection threshold of PADC and its layered structure. We thus turned to the Monte Carlo simulation Geant4-DNA, that has been developed to model radial energy deposition around ion track [38]. From this model, it was possible to reproduce dose distribution that is concordant with the results of Katz model in water. Against these backgrounds, new physical parameters have been proposed by Kusumoto et al. to describe the response of polymeric track detector based on the calculated number of secondary electrons around ion tracks [39,40,59].

*REFIT* was defined as the number density of secondary electrons that traverse a cylindrical surface co-axial with the ion trajectory. Generally, fluence is defined as the number of particles incident on a sphere divided by the cross-sectional area of the sphere. For the Monte Carlo simulations, the number of electrons were counted passing through the surface of a cylinder of a certain radius, because spatially, the tracks of secondary electrons are of cylindrical symmetry around the ion path. The calculations were made with a step of 0.1 nm in radius [39]. The calculated values of *REFIT* at a radius of 1 nm, which is the half of the length of a repeat-unit of PADC, for proton, He and C ions were ranging from 0.15 to 0.3 electrons/nm^2^. These are in agreement with a Multi-hit model that was obtained using low *LET*-radiations. More recently, *REFIT* was applied to the response of PET [59]. Combining two values of *REFIT*_0.5_ (at a radius of 0.5 nm) and *REFIT*_4.0_ (at a radius of 4.0 nm), the detection thresholds of PET for B, C, N and O ions were conclusive. Etchable tracks were formed in PET when *REFIT*_0.5_ is greater than 10 electrons/nm^2^ with keeping 0.8 electrons/nm^2^ as the value of *REFIT*_4.0_. It was suggested that *REFIT* has an advantage to express spatial distribution of secondary electrons around ion tracks by a combination of two or more values at different distance from ion trajectory.

Another proposed physical parameter to describe the response of PADC has been the Number of Interactions with Secondary Electron, *NISE*, which is defined as the number of interactions in a series of cylinders co-axial to the ion trajectory within a certain radius [40]. The description of the response curve improved compared with that expressed using *REFIT*. The removal cross-section for protons and He, C, and Fe ions has been estimated for PADC by the simulation, taking account of the oxidation process [40]. Except from protons with higher energies, the calculation gave concordant results to those from the experiments.

These new physical parameters based on Geant4-DNA have the possibility to give us an epochal change of the study on relationship between the response and latent track structure.

### 3.7. Detection Threshold for C Ion

Etched track detectors have their own detection thresholds [2]. Usually, some cut-off sizes selected by each researcher are used to distinguish etch pits from etch-pit-like round scratches, which are noise in the recorded signal. To make this distinction, it was necessary for us, to make a clear definition on the detection threshold, from the point of view of fundamental study on the latent track structure. The detection thresholds were defined by determining the original points from which the evolution of etch pit starts along the latent track, when the chemical etching was progressing starting from the front surface of each incident ion trajectory. From these points, it was possible to determine the bulk etching or etching duration from the evolution curves of etch pit radius. This value is graphically obtained as the intercept of the evolution trend curve with the horizontal axis [49]. This definition has worked well for PET and polyimide, Kapton, detectors [49,58].

Recently, we found etch pits in PADC sheets, that show unusual growth behavior during chemical etching (Figure 8). The sheets were exposed to 134 MeV/u C ions in form of a stack, providing a set of irradiated samples with different incident energies. The etch pits for the stopping power of 42 eV/nm kept their radii smaller than 1 µm during the etching up to 60 µm. We have observed similar growth trends down to 39 eV/nm. At this stage, we are adopting this value for the detection threshold for C ion.

We think that etchable regions are segmented and isolated from each other along these latent tracks. In other words, the track etching velocity, *V_t_*, is intermittently and slightly greater than the bulk etching velocity, *V_b_*. The formation of etch pit and their round out should be repeated several times during etching treatments. The formerly rounded etch pits became almost invisible under an ordinal optical microscope. We have included these etch pits within the detection side. We are now conducting simulations of the evolution of these “dwarf etch pits”. In the previous section, we mentioned that at least two radiation-sensitive parts should be destroyed in the radial direction of ion tracks to produce any etchable latent tracks in PADC. Similar discussion should be valid for axial direction of latent tracks.

### 3.8. Chemical Criterion for Track Registration Property

The amount of hydroxyl group formed along the ion tracks has been determined for protons and heavy ions [22]. Correlation between the damage density due to the loss of ether and the amount of hydroxyl group was ascertained. In the cases of He, C, Ne, Ar and Fe ions, one hydroxyl was formed when one ether bond was lost in the average. For proton and Xe ion, two hydroxyls are produced in the average when one ether bond was lost. The damage density for loss of ether, $L_{ether}$, is expressed by the next Equation below the stopping power of 800 eV/nm for the first segment as:(21)$$L_{ether}=390{\left(-dE/dx\right)}^{0.67}$$
where $L_{ether}$ is in a unit of scissions/µm [23]. It is possible for us, therefore, to evaluate the formation density of hydroxyl group for each ion, if the stopping power was from the valid range of Equation (21).

The response of the etching track detectors is usually expressed by the reduced etch rate ratio, V−1, where is ratio of the track etching velocity, $V_{t}$, to the bulk etching velocity,$V_{b}$ Namely, $V=V_{t}/V_{b}$ [2]. Figure 9a shows the response of PADC detector for indicating ions as a function of the stopping power [34]. In Figure 9b, identical response data are plotted as a function of the formation density of the hydroxyl group. As far as we know, this work has been the first trial to relate the track response data of PADC against chemical parameters. Notably, we have observed the lack of both data for response and hydroxyl density around the detection thresholds for each ion. However, we can point out the possibility that the formation density of hydroxyl group can work as an alternative criterion for track registration property of PADC [42].

## 4. Conclusions

Modified structure around latent tracks of protons and heavy ions in PADC has been investigated by means of FT-IR spectrometry, covering the stopping powers ranging from 1.2 to 12,000 eV/nm. Details in preparation procedures for the sample films were given. Derivation of chemical damage parameters, i.e., damage density, effective track core radius and radiation chemical yield, from the experimentally obtained removal cross-sections were described. Layered structure in ion tracks in the radial direction was explained referring to the critical dose observed in experiments using low-*LET* radiations, above which the carbonate ester turned to be radiation-sensitive. To examine the validity of proposed multi-hit model for damage formation, the number of secondary electrons was calculated in Geant4-DNA code. As new physical parameters to describe the response of PADC, Radial Electron Fluence around Ion Tracks, *REFIT*, and Number of Interactions by Secondary Electron, *NISE*, were proposed. It was emphasized that a novel epoch has arrived in the study of the relationship between the response and latent track structure based on the comprehensive results from experiments and simulations. Further studies are still necessary on the birth and early growth behavior of etch pits around the detection thresholds. Finally, the formation density of hydroxyl group along ion tracks was proposed as a potential chemical parameter to describe the track registration property of PADC.

We think the methodological and conceptual developments that have been achieved in our studies on PADC will be available for other polymeric materials, including biomolecules [60].

## Figures and Tables

**Figure 1 polymers-13-02665-f001:**
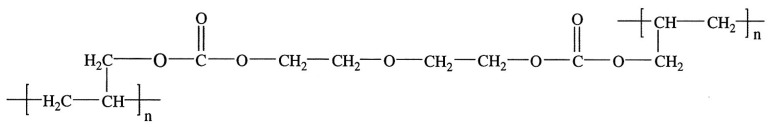
A repeat-unit of PADC.

**Figure 2 polymers-13-02665-f002:**
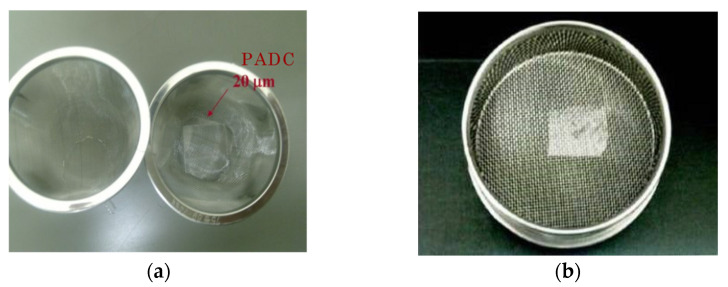
Photographs of PADC films during the thinning processes by chemical etching. The size of PADC films is 20 × 20 mm^2^. (**a**) PADC film with a thickness of 20 µm on stainless sieve, in the second step; (**b**) PADC film with a thickness of 20 µm in the final stage of etching on a stainless sieve with a flat base [12].

**Figure 3 polymers-13-02665-f003:**
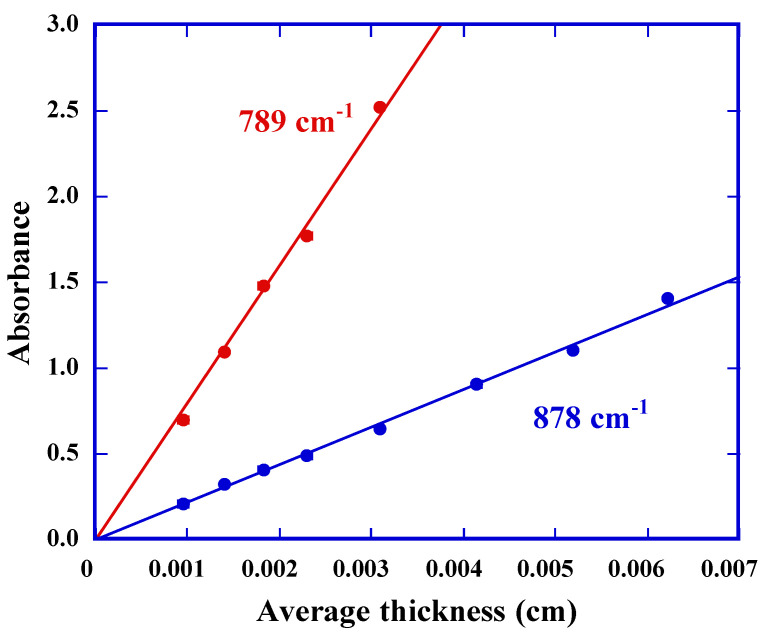
Absorbance of IR peaks of CH groups at indicating wavenumber, as a function of the average film thickness measured by a micrometer [12,15].

**Figure 4 polymers-13-02665-f004:**
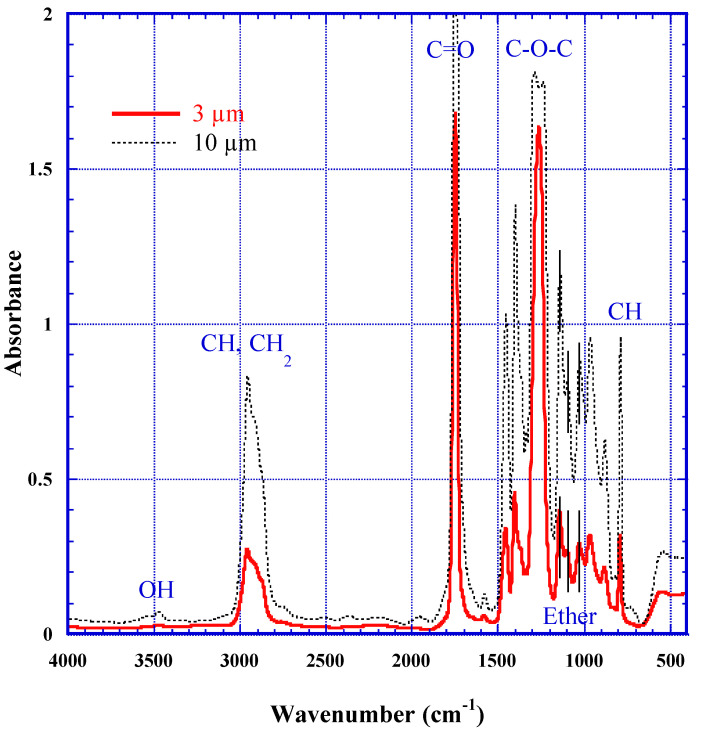
FT-IR spectra of PADC films with thicknesses of 3 and 10 µm.

**Figure 5 polymers-13-02665-f005:**
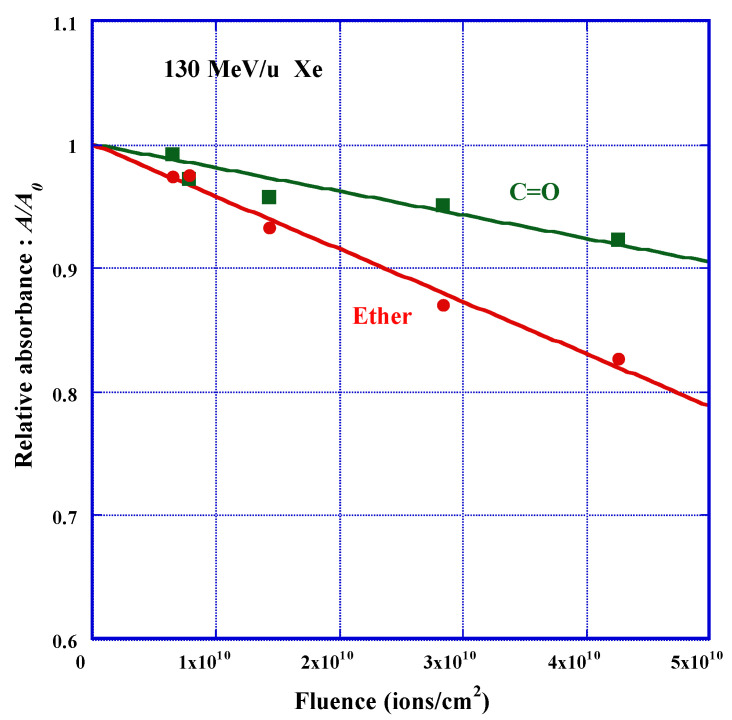
Reduction of the relative absorbance of ether and carbonyl in PADC against the fluence of 130 MeV/u Xe ions.

**Figure 6 polymers-13-02665-f006:**
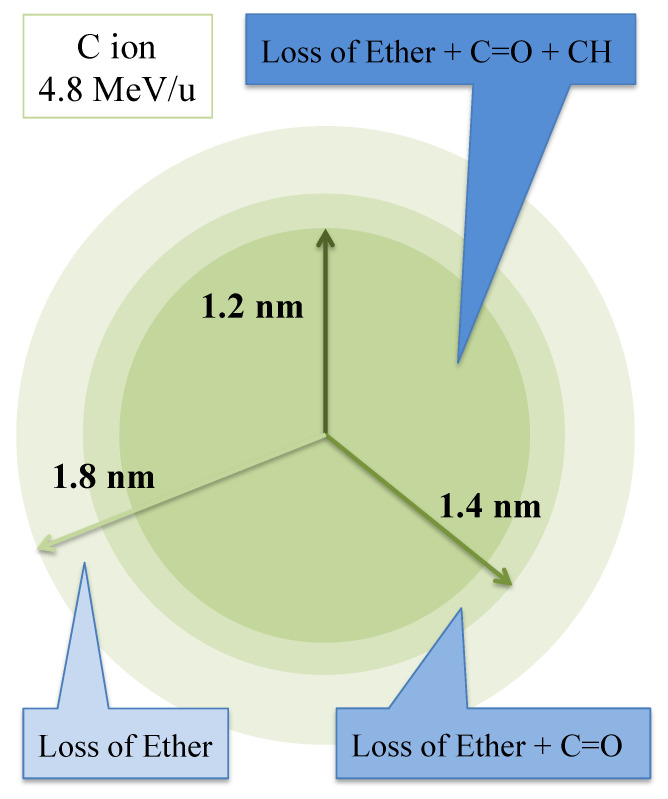
Layered structure of C ion track with an energy of 4.8 MeV/u.

**Figure 7 polymers-13-02665-f007:**
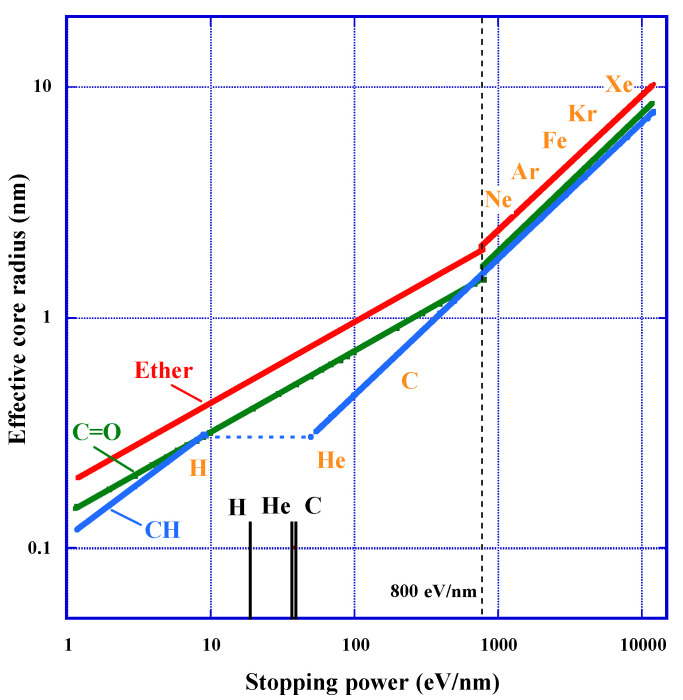
Effective track core radius for losses of ether, carbonate ester and CH groups.

**Figure 8 polymers-13-02665-f008:**
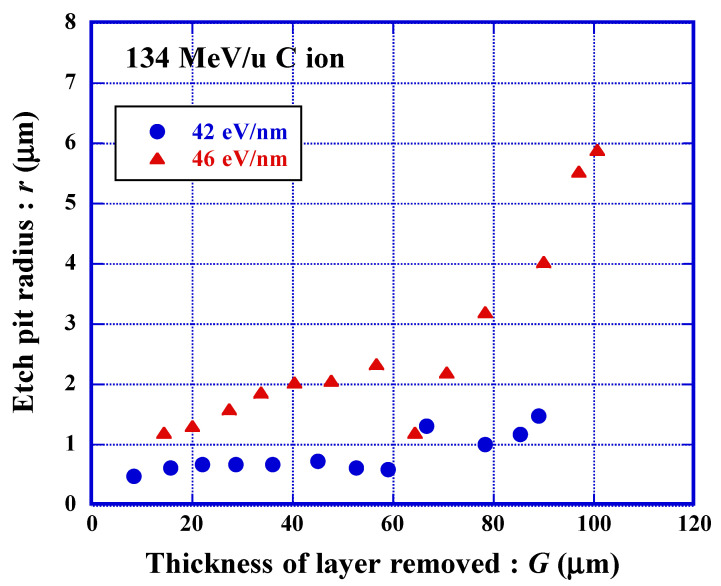
Changes in observed etch pit radius of C ions with indicating stopping powers against the thickness of layer removed.

**Figure 9 polymers-13-02665-f009:**
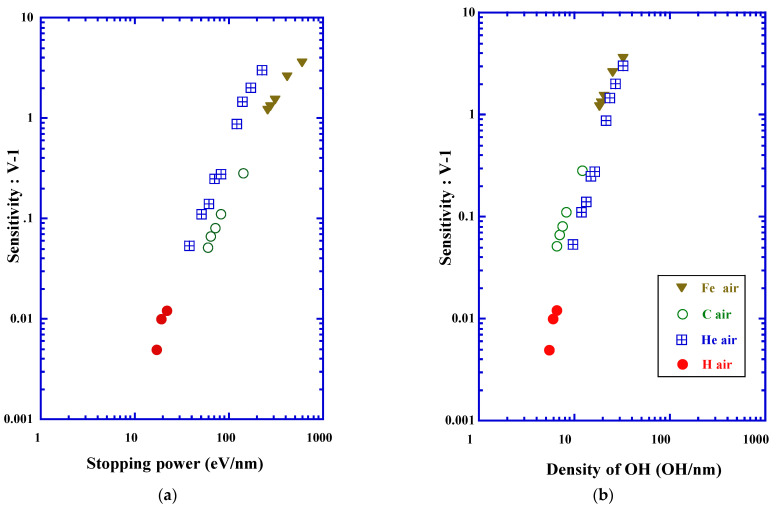
Track response data (**a**) as a function of the stopping power and, (**b**) as a function of the formation density of hydroxyl group.

## Data Availability

The data presented in this study are available on request from the corresponding author.

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
