# Peer review of "Methodological and Conceptual Progresses in Studies on the Latent Tracks in PADC"

_polymers, 2021, doi:10.3390/polym13162665_

Round 1

Reviewer 1 Report

Tomoya Yamauchi et al. provide a comprehensive review on their recent research progress in studying the latent tracks in PADC, especially on the FITR measurements on thinned films, a new explanation on the damage mechanism, and better correlated physical criteria to describe the detection threshold. This timely review will advance our understanding of the latent track formation in polymer membranes. I would recommend its publication. However, the authors should carefully correct some typos. For example, Line 199, add "is" in "which relatively independent.."; Equation 12 and 13 show strange symbols. 

Author Response

Reply to Reviewer Report

At the first, thank you for reviewing our manuscript. 

"Tomoya Yamauchi et al. provide a comprehensive review on their recent research progress in studying the latent tracks in PADC, especially on the FITR measurements on thinned films, a new explanation on the damage mechanism, and better correlated physical criteria to describe the detection threshold. This timely review will advance our understanding of the latent track formation in polymer membranes. I would recommend its publication. However, the authors should carefully correct some typos. For example, Line 199, add "is" in "which relatively independent.."; Equation 12 and 13 show strange symbols. "

We have modified the sentence (line 199), as “Usually, we chose to measure the third peak, which is relatively independent from others.”

About the strange symbols, we would like you open the attached pdf.

We think any strange symbols will not appear in printed matters.

We have corrected confused equation numbers, caused by our careless mistakes.

Reviewer 2 Report

Review of manuscript Manuscript Number: Manuscript ID: polymers-1316182 for Journal Polymers

Title: Methodological and Conceptual Progresses in Studies on the Latent Tracks in PADC

Authors: Tomoya Yamauchi, Masato Kanasaki and Rémi Barillon

Report

This paper deals with modified track structure and track formation processes in poly (allyl diglycol carbonate), PADC, used as a sensitive etched track detector. The authors have been investigated the track structures of protons and heavy ions (He, C, Ne, Ar, Fe, Kr and Xe) and have been examined them by means of FT-IR spectrometry, covering the stopping power region between 1.2 to 12,000 eV/nm. Through a set of experiments on low-LET radiations such as, gamma ray -, multi-step damage process by electron hits was confirmed in the radiation sensitive parts of the PADC.  

The authors have been evaluated the number of secondary electrons around tracks, and they have been performed a series of numerical simulations with Geant4-DNA. The authors have been proposed new physical criterions to describe the detection thresholds, and present an issue of the definition of detection threshold for semi-relativistic C ions.

I have no important recommendations to modify and/or change the text of presentation except the minor corrections:

  1. Arrows are overlapped in equations 2 and 3.
  2. Page 8, equation 13 is not clear or misprint.
  3. Equation 12 in pages 8 and 9 have the same number, please check.
  4. Same embed as before, eq. 13, page 10, please check.
  5. Page 10, Eqs. 13 1nd 14, rether, please give a labeling for differentiation.
  6. Same embed as before, Eqs. 15 and 16, rC=O, please give a labeling for differentiation.
  7. Same embed as before rCH, Eq. 17, please give a labeling for differentiation and check the number of equations.

This is a very good piece of work and it is a good contribution to polymer and nuclear track sciences. Thus, I do recommend this manuscript for publication in Polymers journal after minor revision.   

Author Response

Thank you for reviewing our manuscript. Our responses are written in the attached file.

This manuscript is a resubmission of an earlier submission. The following is a list of the peer review reports and author responses from that submission.

Round 1

Reviewer 1 Report

Dear Editor,

The manuscript “Studies on the Latent Tracks in PADC” proposes to review the data on track formation on poly allyl diglycol carbonate (PADC), which is a well-known detector used in radon and neutron dosimetry, as well as in space dosimetry.

Although the topic is of general relevance to better understand this material, I find the manuscript very confusing, particularly for a review article. Some of the problems are listed below.

  1. I find the entire structure of the manuscript confusing. The structure of the paper oscillates between a review paper and a regular contribution and it is not easy to understand what is been reported for the first time in this manuscript, and what has already been published before. The manuscript needs to be completely re-structured.
  2. The introduction needs to review the state-of-the-art prior to the work from the authors. Section 1.2 is confusing here, since this is the topic of the present review. Also, the first sentence of the introduction is superfluous. The statement of purpose should be at the end of the introduction.
  3. The “Materials and Methods” Section should make more clear the experimental procedure used. It is not clear which samples were irradiated and if the samples were further etched after irradiation or not. The section should also be better structured, with sub-sections (sample preparation, FT-IR measurments, irradiations, data analysis, etc.). Right now, different topics are mixed in the same paragraph. Currently results are being presented in the “Materials and Methods” section, which is not appropriate.
  4. The “Results and Discussion” should explain better the motivation of each study and the relevance of the results. Right now one cannot understand the reason each study was performed. It is also not clear if the results are new or already published before. \
  5. Many sentences are not clear or not well justified: E.g.: “It is defined as the radius in which the considered chemical bonds are lost as in the original one.” This sentence is incomprehensible.

Other minor points:

  1. The authors indicate in the introduction that with the analysis of the etch-pit one can identify the nuclear species and evaluate the incident energy. I don’t believe this is correct. One can only determine the restricted LET. Different particles/energy combinations with the same restricted LET will not be differentiated by the PADC.
  2. There are many problems with rendering the symbols in the equations, which make them incomprehensible.
  3. Throughout the manuscript, the present perfect (e.g. “We have calculated…”) is not necessary: simple past is sufficient (“We calculated…”).

Until the manuscript is better structured and better written, it is impossible to comment more specifically on the science. I think there is the potential for this manuscript to be a significant contribution to the field by summarizing the apparently extensive studies on this material. Nevertheless, unless the technical writing and the overall explanation of the problem, experiments and results are considerably improved, my recommendation is for the rejection of the manuscript.

Author Response

Dear Reviewer,

Thank you very much for reviewing our contribution. Please find our attached file.

Reviewer 2 Report

Tomoya Yamauchi et al. provide a comprehensive and timely review on their last decade of studies on latent track structures and track formation process in PADC (CR39). Their studies are mainly based on quantitative FT-IR spectrometry measurements on the PADC films irradiated with a series of different ions with different LET values.  They calculated the effective track core radius, rt, to show the radial size of the radiation damage and showed the layer structure of ions. They analyzed the dependence of rt on dE/dx and identified the detection threshold for different functional groups.  They further ran Geant4-DNA simulations to compute the distribution of the secondary electrons around tracks and proposed the physical criterion and chemical criterion for the detection thresholds. This review summarizes the recent progress on the latent track in PADC and the other polymer membranes and discusses the future study directions. It deepens our understanding of the formation mechanism of the latent tracks and will facilitate the design of new sensitive etched track detectors. I would recommend its publications if it could address the following questions. 

1. P7-8, the authors applied the Beer-Lambert low with the assumption that the latent tracks are not overlapped. It would be better that they discuss the criteria in which such an assumption is valid. 

2, P9, please explain a little bit more on why "This is one of the great features of PADC".

3. P9, it would be helpful to show explicitly the relationship between G and rbefore the discussion of the relationship between these two values and dE/dx late. 

4. There are still some typos and grammar mistakes. For example, 
Line 72, "ultraviolet, (UV)"
Line152, "The size of PADC films are"
Eq (3) the subscript of A should be 878 not 787
Line 284, "were occurred"
Eq (12), "p" should be the symbol pi
Eq (13), strange symbols on the two sides of dE/dx
Line 366, "vales" should be "values"

Author Response

(The authors gave the same response as above.)

Reviewer 3 Report

Review of manuscript Manuscript Number: Manuscript ID: polymers-1127575 for Journal Polymers

Title: Studies on the Latent Tracks in PADC

Authors: Tomoya Yamauchi, Masato Kanasaki and Rémi Barillon

Report

This review paper deals with latent track structures and track formation process in poly (allyl diglycol carbonate), PADC, used as a sensitive etched track detector. During the last two decades, the Japanese -French team studied the track structures of protons and heavy ions (He, C, Ne, 11 Ar, Fe, Kr and Xe) by means of FT-IR spectrometry, covering the stopping power region. Through a set of experiments on low-LET radiations  such as, gamma ray -, multi-step damage process by electron hits was confirmed in the radiation sensitive parts of the PADC.  

A large amount of precise and interesting measurements previously published in good profile journals has been collected. Anyhow the data has been analyzed, organized and ordered in a very good manner. This review paper has an excellent knowledge to warrant of publication in journal of Polymers after revision.

  1. Page 2, paragraph from lines 54-59. Please move it to the acknowledgments.
  2. Page 2, line 81, please correct "tacks" to "tracks".
  3. The caption of Fig. 2, does the thickness film is 20 µm in both (a) and (b). Please clarify it.
  4. Page 3, lines 96-100, reference [14] is not adequate, and some research papers on similar measurements were carried out by several groups, please give more citations.  
  5. The caption of Fig. 3, just, I wonder do the authors have measured the film thickness by micrometer in cm or in µm and what is the uncertainty in cm for these measurements. If you are used µm in your measurements, if you present your data in µm, I think it gives more clarification. Please correct it.
  6. Page 4, equations 2 and 3, arrows interfere with numbers. Please correct.
  7. Page 6, Fig. 6 and related paragraph in the text, it is so nice to measure FTIR spectra of PADC film with thicknesses of 3 to 10 µm to prevent the saturation of the strong peaks. Just I wonder the authors handle these films with thickness 3 µm, even 10 µm is not so easy. In my opinion and maybe I am not right, but I can't satisfied about the production and handling of 3 µm thickness in spite of I know these data are already published since long time ago. Please clarify.
  8. Page 6, line 232, and please correct "kG" to "kGy".
  9. Page 8, equation 13 is not clear, may be something wrong during conversion from word to pdf.
  10. Page 10, line 352, the authors said "we have calculated the experimental formula……". May be mean, "We have used the empirical formula based on experimental data". Please rephrase this sentence again.
  11. Page 11 lines 384-385, "It is noteworthy that the G values for the protons are remarkably higher than those for the heavy ions, which is comparable to that for gamma rays [23]". It is strange to find that the G-values of heavy ions are comparable to that for gamma rays and lower than those of protons. Please give some explanation about the dose and the effects of dose rate on G values.   
  12. Page 14, the caption of Fig. 9 "Track response data as a function of the stopping power (a) and the formation density of hydroxyl group (b)". Please modify this caption as"Track response data (a) as a function of the stopping power and, (b) as a function of the formation density of hydroxyl group.

This review paper is an excellent piece of work and I would say that it is highly significant contributions to polymer science and to nuclear track physics and chemistry. Thus, I do recommend this manuscript for publication in Polymers journal after revision.    

Author Response

Dear Reviewer.

Thank you very much for reviewing our contribution. Please find our attached file.

Round 2

Reviewer 1 Report

I appreciate the efforts by the authors, but in my opinion the main difficulty has not been addressed.

1. For a review paper, the manuscript is not sufficiently clear for a non-expert in the field (as myself) to understand the advances described in the paper.

2. For a paper with new results, the new results are mixed with published results and it is difficult to distinguish between them.

Because of that, I consider extremely difficult to evaluate the merits of the manuscript. The study is certainly worth and the results may be significant, but as it stands I cannot say that this is a good review paper or that there is good new scientific contribution. I particularly discourage the mixture of review paper and new data, unless the new data is used for illustrative reasons.

Although I cannot recommend publication of the manuscript, I would encourage the Editor to find for another expert that may be able to understand better the science and decide if such paper format (mix of review and new data) makes sense.

Author Response

Thank you very much for reviewing our contribution, again.

We submit our notes to the reviewer.

Reviewer 3 Report

The paper is an excellent piece of work and I do accept it for publication in Polymers Journal

Author Response

Thank you very much for reviewing our contribution, again.

We are really encouraged by your constructive comments.